# Three-dimensional wind profiles using a stabilized shipborne cloud radar in wind profiler mode

Alain Protat[1], Ian McRobert[2]

[1] Australian Bureau of Meteorology, Melbourne, Victoria, Australia

[2] Engineering and Technology Program, CSIRO Oceans and Atmosphere, Hobart, Tasmania, Australia

*Correspondence to*: Alain Protat (alain.protat@bom.gov.au)

**Abstract.**

In this study, a shipborne 95 GHz Doppler cloud radar mounted on a stabilized platform is used to retrieve vertical profiles of three-dimensional (3D) winds by sequentially pointing the stabilized platform in different directions. A specific challenge is
that the maximum angle off zenith is 8°, which implies that the projection of the horizontal wind components onto the radar beam directions is a small component of Doppler velocity in most cases. A variational 3D wind retrieval technique is then described, allowing for 1-minute 3D wind profiles to be retrieved. Statistical comparisons with 3-hourly radiosonde launches from the ship indicate that horizontal wind profiles can be obtained from such cloud radar observations at small off-zenith angles with biases less than 0.2 ms$^{-1}$ and standard deviations of differences with radiosonde winds less than 2.5 ms$^{-1}$.

## 1 Introduction

Vertically-pointing Doppler cloud radars provide unique observations to better understand the interactions between dynamics and microphysics in clouds and light precipitation, and cloud radiative forcing. Doppler cloud radars are also extensively used to evaluate satellite products and the representation of cloud and precipitation properties in models. The focus of studies has been on retrieving the microphysical properties of clouds from cloud radars (e.g., Matrosov et al. 2002; Mace et
al. 2002; Delanoë et al. 2007), lidars (e.g., Heymsfield et al. 2005), or cloud radar – lidar combination (Wang and Sassen 2002; Okamoto et al. 2003; Tinel et al. 2005; Delanoë and Hogan 2008, Deng et al. 2010). The next challenge to better understand the interactions between dynamics and cloud microphysics is to characterize the dynamical context of these cloud microphysical observations, including horizontal winds, vertical wind shear, and entrainment processes within and at the boundaries of clouds.


Scanning cloud radars were recently developed to describe clouds in three dimensions (3D) from ground-based observatories. However, scanning strategies need to be adapted to characterize 3D wind profiles at high vertical resolution within clouds using such measurements, as typical scanning strategies currently focus on describing the morphological structure using Plan Position Indicator (PPI, scanning in azimuth at successive constant elevations) or Range Height Indicator (RHI, scanning different elevations at constant azimuth) scanning sequences, which does not allow for the 3D wind profiles

30 to be retrieved without stringent assumptions (e.g., linearity of the wind components). UHF and VHF profilers provide such 3D wind profiles in clear-air and precipitation from Bragg and Rayleigh scattering, respectively, using the so-called "profiler mode" that consists of alternating vertical pointing with off-zenith pointing by about 15-20° in two perpendicular directions (North and East for instance). The only issue with UHF and VHF wind profilers is that they lack the sensitivity to detect thin non-precipitating clouds.

35 In this study, we report on a pilot study using a shipborne cloud radar on a stabilized platform to derive high-resolution vertical profiles of 3D wind. The idea is to use the stabilized platform to point the cloud radar in a series of different directions. However, when the cloud radar is used on the Marine National Facility (MNF) Research Vessel (RV) *Investigator*, the maximum angle off zenith that can be pointed safely at with the stabilized platform is ±8° in pitch and roll directions due to the size of the aperture on the container roof. This is smaller than typical angles of ±15-20° used for wind profilers. The main

40 objective of this pilot study is to assess whether high-quality 3D winds can be retrieved from such small angles off zenith. As this study was conducted during a major field experiment, the Years of the Maritime Continent – Australia (YMCA), radiosondes were launched every three hours from the ship, allowing for a quantitative evaluation of the retrieved cloud radar horizontal wind profiles. In section 2, we briefly describe the cloud radar, the stabilized platform and implemented scanning sequences. In section 3, we illustrate the retrieval with case studies and conduct a statistical evaluation of the cloud radar

45 horizontal winds against radiosonde measurements. Conclusions are finally given in section 4.

## 2 Description of the pilot study

In this section we briefly describe the cloud radar, the stabilized platform and implemented sampling strategies, and the 3D wind retrieval technique.

### 2.1 The BASTA Doppler cloud radar

50 The research-grade BASTA Doppler cloud radar is described in detail in Delanoë et al. (2016). It is a frequency-modulated continuous wave (FMCW) radar operating at a frequency of 95 GHz. The radar uses two Cassegrain dishes (60 cm in diameter) and all the electronic components are installed in a pressurized and insulated box. The data acquisition and processing are done using a field-programmable gate array (FPGA). This cloud radar uses a low-power solid-state transmitter (0.5W) and estimates both reflectivity and Doppler velocity using the pulse-pair processing technique with 2048 samples,

55 allowing for high Doppler measurement accuracy (better than 1 cms$^{-1}$ Doppler spectral accuracy). During the YMCA field experiment, a 12 seconds sequence split into four successive modes (each mode with an acquisition and processing time of 3 seconds) was designed to capture both low-level clouds and light precipitation with high vertical resolution and tropical cirrus clouds with high sensitivity. The respective vertical resolutions of these four modes are 12.5m, 25m, 100m (moderate sensitivity), and 100m (higher sensitivity but shorter Nyquist velocity). The approximate minimum detectable signal of these

60 four modes is -28, -34, -40, and -43 dBZ at 1 km range, respectively. This sensitivity is lower than that reported in Delanoë et al. (2016), due to current issues with the antenna alignment. This lower sensitivity is not detrimental to our pilot study.

## 2.2 The *RV Investigator* stabilized platform and sampling strategies

When used on *RV Investigator*, the BASTA cloud radar is mounted on a stabilized platform inside an air-conditioned container, ensuring resilient operations in harsh environments such as the Southern Ocean, Antarctica and the Tropics. The stabilized platform design is described in detail in Filisetti et al. (2017) and follows the design from Moran et al. (2012). It has been recently demonstrated that vertical stabilization to better than $0.2°$ can be achieved with this platform for sea states up to 6. In this shipborne configuration, the plexiglass dome of the BASTA cloud radar is removed and replaced by a bigger one mounted directly on the container roof. Due to the size of the aperture made on the container roof and the requirement to minimize contaminations of the signal by multiple reflections on the metallic structure inside the container, the top of the cloud radar is lifted very close to the dome. This configuration limits the possible rotation in pitch and roll directions to about $12°$ from the vertical of the container. Our experience from the Southern Ocean high seas is that with the anti-roll system of *RV Investigator* this value of $12°$ has been exceeded less than 1% of the time.

The baseline operating mode in earlier deployments was the "vertical mode", where the instrument is stabilized to point vertically all the time. For this pilot study we have developed an additional mode, referred to as the "profiler mode" in the following, which consists of a 120 seconds sequence with 15 seconds spent at the following 8 pointing angles: vertical, $+8°$ pitch, vertical, $+8°$ roll, vertical, $-8°$ pitch, vertical, $-8°$ roll. With such a sequence, we still retain a high temporal resolution for the vertical observations while being able to retrieve 1-minute 3D wind profiles from any of four successive pointing angles. The rationale for using positive and negative pointing angles is to assess whether the same 3D wind profiles can be derived from these different combinations of angles. Note that the time (about 1.5s) required to move from one angle to the next is included in the 15 seconds. The selected time interval of 15 seconds is a trade-off to make sure that we are collecting data from all four radar modes, which takes 12 seconds, for each pointing direction while still retaining a small time interval (1 minute) between retrieved 3D wind profiles. The $8°$ angle selected for this mode is also a trade-off between allowing enough projection of the horizontal wind components onto the radar beams off zenith, and the need to stabilize the instrument in that direction accurately. Using $8°$ means that we can only stabilize the instrument for motions less than about $4°$. Although this will present a challenge in rough seas such as over the Southern Ocean, such motion was never encountered during the YMCA experiment.

## 2.3 The 3D wind retrieval technique

When operating in vertical mode, Doppler velocities are simply corrected for heave rates (the vertical component of ship speed) using the 10 Hz ship positioning system data (same as in Moran et al. 2012). When operating in modes with off zenith pointing, Doppler velocities need to be corrected for both heave rates and ship horizontal speed. During YMCA, we mostly stayed on station and heave was very low. As a result, Doppler corrections very rarely exceeded absolute values of 0.2 ms$^{-1}$ (not shown). However, it will not be the case for future deployments. Therefore, below we develop the full set of equations for the 3D wind retrieval including all corrections.

A variational 3D wind retrieval has been adapted for the profiler mode sampling strategy from the dual-Doppler weather radar technique of Protat and Zawadzki (1999). The three "control variables", i.e., the quantities to be retrieved, are the zonal (eastward) horizontal wind component $V_X$ *(nt, nz)*, the meridional (northward) horizontal wind component $V_Y$ *(nt, nz)* and $V_Z$ *(nt, nz)* = $W$ *(nt, nz)* + $V_T$ *(nt, nz)*, where $W$ *(nt, nz)* is the vertical wind component and $V_T$ *(nt, nz)* is the terminal fall velocity of hydrometeors, *nt* is the number of time steps per retrieval day and *nz* is the number of vertical levels for the vertical profiles. The *nt* and *nz* parameters can be adjusted for different applications. When operating in profiler mode instead of the traditional weather radar PPI sampling, which mostly involves low elevation angles above the horizontal plane, the anelastic airmass continuity equation and the constraint that the vertical air velocity at ground is nil are not needed as part of the retrieval process. As a result, only the Doppler velocity constraint from the set of constraints used in Protat and Zawadzki (1999) is used, and includes all pointing angles to retrieve the vertical profiles of 3D wind. As a result, the cost function to be minimized can be simply written as:

$$J = \sum_{i=0}^{nt} \sum_{k=0}^{nz} (V_R(i,k) - V_R'(i,k))^2 \qquad (1)$$

where

$$V_R'(i,k) = \left(V_X(i,k) - U_{ship}(i)\right) \cos\left(az(i)\right) \cos\left(el(i)\right) + \left(V_Y(i,k) - V_{ship}(i)\right) \sin\left(az(i)\right) \cos\left(el(i)\right) \qquad (2)$$
$$+ \left(W(i,k) + V_T(i,k) - W_{ship}(i)\right) \sin\left(el(i)\right)$$

with $V_R(i,k)$ the measured Doppler velocities, $V_R'(i,k)$ the theoretical Doppler velocities matched to the observed Doppler velocities $V_R(i,k)$ as part of the minimization process, $(U_{ship}(i), V_{ship}(i), W_{ship}(i))$ the three components of the ship speed producing apparent Doppler velocity in the cloud radar measurements that need to be subtracted to the theoretical Doppler velocities, and $(az(i), el(i))$ the azimuth angle of each radar beam with respect to the east (positive counter-clockwise) and the elevation angle of each radar beam with respect to the horizontal (positive upwards).

The different steps of the procedure to minimize the cost function *J* can be summarized as follows: 1) make an initial guess of the control variables ($V_X$, $V_Y$, $V_Z$) – we use zero by default ; 2) calculate the gradient of the cost function with respect to the control variables (as explained in Protat and Zawadzki 1999); 3) exit if the predefined convergence criterion is met; otherwise, 4) calculate a new guess of ($V_X$, $V_Y$, $V_Z$) using the conjugate–gradient method (Powell 1977); and 5) return to step 2 for a new iteration using this new guess until the convergence criterion is met. Once $V_Z$ is obtained, previous studies have shown that its two components, $W$ and $V_T$, can be separated using statistical approaches relating reflectivity to $V_T$ (see description of different possible techniques and expected performance in Protat and Williams, 2011). However, because there is no reference observation to evaluate this in our dataset, this separation of $W$ and $V_T$ has not been included in the present analysis.

## 3 Results

The pilot study to test the new wind profiler mode was conducted during the YMCA field experiment (12 November 2019 to 17 December 2019). Large-scale conditions during the experiment were not favourable for the development of offshore propagating mesoscale convective systems and associated cloud anvils and tropical cirrus layers. Nevertheless, a variety of
cloud cover types has been sampled over that period.  In this section, we present results obtained for four very different cases to illustrate the capability to retrieve 3D winds from the cloud radar in profiler mode in different situations. The first case is a stratiform precipitation case that developed on 23-24/11/2019 within a horizontal flow characterized by multiple vertical wind shear layers. This case was over the ship for about seven hours. The second case is a shallow cumulus congestus case that developed in the evening of 24/11/2019 in a strong low-level vertical wind shear environment, as measured by the soundings.
The third and fourth cases consist of an altostratus and a tropical cirrus outflow, respectively. In both cases, sampled clouds were detrained from surrounding deep convective activity on 04/12/2019. Then, we present a statistical analysis of comparisons with radiosonde winds using all retrieved horizontal wind profiles from 22/11/2019 to 04/12/2019.

Two types of measurements are used in this study to compare with retrieved horizontal winds, both bringing complementary insights. Note that we do not have independent measurements to assess the vertical wind component. However,
this component is directly measured with the current sampling strategy, so it can be assumed accurate to within measurement and Doppler corrections uncertainties. The first measurements are ship-level horizontal winds measured at 24m height on the front mast by two automatic weather stations. Comparisons are made with the first valid radar range bin where winds can be retrieved (usually about 100m height). The limitations of such comparisons are the difference in heights of the measurements, and the fact that it does not allow for an assessment of the full vertical profiles, only those situations when low clouds are
present. Since the final post-processed version of these weather station winds have not been produced yet, we only use these for qualitative illustration. The second type of measurements is soundings (Vaisala RS41-SGP radiosondes), which were launched every three hours from the ship during YMCA. This second source of validation has the major advantage of providing full vertical profiles of horizontal winds surrounding the cloud radar retrievals. However, balloons take about one hour to reach the tropopause in the Tropics and can drift by tens of kilometres from the initial launch location over that period. As a result,
the main potential issue when comparing radiosonde and retrieved horizontal winds is that differences obtained include an unknown contribution from the true spatial and temporal variability of the wind components, including that produced by internal cloud dynamics, in addition to the retrieval errors.

These advantages and limitations have informed the way comparisons are made for case studies and on a more statistical basis in this section. Low-level time series of horizontal wind components have been averaged using the two wind
measurements from the weather stations. These are displayed on the vertical cross-sections of retrieved winds at altitude zero with the same colour code. For radiosonde comparisons using case studies, we have selected individual four-hour periods of interest bounded by two radiosonde launches, and we assume that the two radiosonde wind profiles can be used as a proxy for the spatial and temporal variability of the horizontal wind components. The rationale for choosing four-hour periods is because

this is the time difference between the time of launch of the first sounding and the arrival time of the second sounding at the tropopause. For each four-hour period, a joint horizontal wind – height distribution of the retrieved horizontal wind components is produced to compare the variability of these wind components with that derived from the two soundings. Subsequently, if the spread of wind retrievals (i.e., variability) within the four-hour period is bounded by the two radiosonde wind profiles, we qualitatively label the comparison as a "good agreement", because the retrievals are within the natural variability captured by the two soundings, Although these comparisons are more of a qualitative nature, they allow for a good visual assessment of the retrieved horizontal wind profiles.

Figures 1-3 show results obtained for the stratiform precipitation case. The sum of vertical air motion and terminal fall speed of hydrometeors (top panel of Fig. 1) is characterized by an expected sharp transition from downward vertical motions in the -2 to 0 ms$^{-1}$ range in ice phase above the melting layer height (around 4.5 km), to values below -4 ms$^{-1}$ in liquid phase, below the melting layer height. Stratiform regions are generally characterized by relatively small vertical air motions, rarely exceeding 0.5 ms$^{-1}$ (e.g. Protat and Williams, 2011, in the same Darwin region). As a result, the sum is generally dominated by terminal fall speed. Layers of enhanced downward motions in ice phase closer to the melting layer result from the aggregation of ice crystals producing bigger particles as they fall within the stratiform region, as documented in numerous studies. Values of near zero vertical motions are also found near cloud top, which is also the expected signature of much smaller ice crystals falling at a much lower speed. The two retrieved horizontal wind components (middle and bottom panels of Fig. 1) are characterized by long-lasting structures of higher easterly and south-westerly winds just below (2 to 4 km height) and just above (6 to 8 km height) the melting layer, respectively. An upper-level south-westerly jet is also clearly visible on the retrieval (above 10 km height). Qualitative validation of the low-level winds is shown in Fig. 2. Except for a short period after 0000 UTC where some clear differences are observed, the agreement between ship horizontal winds and retrieved winds is good, with subtle changes in horizontal wind speed and direction picked up in the retrieval. Looking more closely at the ship time series, it appears that this short period is characterized by very large differences in excess of 10 ms$^{-1}$ between the port and starboard weather station estimates, with our retrieved values being closer to one of the estimates.

A statistical comparison of the retrieved vertical profiles of the horizontal wind components with the radiosonde observations is shown in Fig. 3 (radiosondes are also superimposed to retrievals in Fig. 1) for the two 4-hours periods depicted in Fig.1. For each period we have two radiosonde profiles to compare with. Comparing profiles from the two radiosondes indicates that there is substantial variability of the zonal wind component above 6 km height for the first period (top panels of Fig. 3). The retrieved horizontal winds closely match the radiosonde profiles below 5 km height, and the agreement is also very good in the upper levels, with the two radiosonde observations generally bounding the retrieved horizontal wind distributions. This good qualtitative agreement for the stratiform precipitation case holds true for different types of cloud cover we analyzed, including a cumulus congestus case characterized by high vertical wind shear (Figs. 4 and 5), an altostratus case in a very light wind environment (Figs. 6 and 7), and a tropical cirrus case embedded in a north-westerly jet (Figs. 6 and 7). Comparisons with ship-level weather stations and with the radiosondes bounding the congestus case (Figs. 4 and 5) indicate that the near-surface winds are well reproduced, and the strong vertical wind shear in the low levels (about 5 10$^{-3}$ s$^{-1}$ in the 0-

3 km layer) is accurately captured by the retrieval. We note some large differences with the radiosonde on the $V_Y$ component at cloud top, which does not necessarily mean that the retrieval has failed, as tops of congestus clouds are notoriously turbulent. Therefore, these differences could be due to the internal convective-scale dynamics not captured by soundings obtained in clear-air. The altocumulus and cirrus cases are characterized by a much lower temporal variability of the wind components (see for instance the small difference between estimates from the two soundings for the cirrus case, Fig. 7b). Accordingly, the frequency distributions of retrieved horizontal winds over four hours is much narrower than within the other cases, and the peaks of the distributions align well with the radiosonde measurements for those two cases.

The case studies presented previously do not provide a quantitative evaluation of the retrieved horizontal winds. As explained earlier, the main challenge of these comparisons with soundings is that differences include two components: the errors from the retrieval, which is what we would like to characterize, and the spatial and temporal variability of the true horizontal wind components captured by the soundings as the balloons drift away from the ship location during the hour it takes from the balloon to go from ground to the tropopause. Despite these challenges, we can exploit the fact that the second term should vanish when the distance and the time difference between the retrieved and measured horizontal winds are small. In order to investigate these effects and quantify errors in our horizontal wind estimates as accurately as possible, all comparison points for the 22/11/2019-04/12/2019 period are binned as a function of distance (every 0.5 km from 0 to 10 km) and absolute time difference (every 2 minutes from 0 to 60 minutes) between sounding measurements and retrieval points. The statistical frequency distribution of errors (normalized to 100% in each bin), bias, and standard deviation of the differences are presented in Fig. 8 for distance and Fig. 9 for absolute time difference. Fig. 8 indicates that quantitative comparisons can be made between soundings and retrievals up to about 7 km distance (with fluctuations of the bias and standard deviation of about 1 and 1.5 ms$^{-1}$, respectively). However, it appears clearly that the frequency distribution of errors is more peaked for distances below 1 km, showing as expected some impact at relatively short distance (from 1 km and more) of the spatial variability on the accuracy of our error estimates. The statistical analysis of all comparison points at distances less than 0.5 km (6.4% of all points), which is our best estimate of the retrieval errors since they include the smallest possible contribution from the true spatial variability, indicate that virtually unbiased (less than 0.2 ms$^{-1}$) estimates of the two horizontal wind components are obtained, with a standard deviation of the error of 2.4 ms$^{-1}$.

Reorganizing the comparison points using bins of absolute time difference between soundings and retrieval points (Fig. 9) yield similar results to those obtained when binning using distance. From the samples used to produce Fig. 9, we obtain that the temporal evolution of the horizontal wind components has a large impact on our error estimates beyond time differences of 30 minutes, but a reasonably small effect on our error estimates for absolute time differences up to about 15-20 minutes (a variability of about 1 ms$^{-1}$ for the bias and standard deviation of the error), with the distribution of errors getting broader for absolute time differences greater than about 15-20 minutes. As was observed for the analysis as a function of distance, the frequency distribution of errors is much more peaked for times less than about 6 minutes, which is therefore where we expect minimal contamination of our error estimates due to temporal variability. It is noteworthy that the frequency distribution for small distances is much more peaked than that for short time differences. This suggests that the temporal

variability of the horizontal wind components has more impact on our error estimates than the spatial variability. The statistical analysis of all comparison points with less than 2 minutes of absolute time difference (8.5% of all points) confirms the small bias of the two horizontal wind components (less than 0.4 ms$^{-1}$) that was obtained using distances smaller than 0.5 km, and a

230    similar but slightly higher standard deviation of the error (2.9 ms$^{-1}$) than when binning using distance. Using both distances less than 0.5 km and absolute time differences less than 2 minutes (3% of all comparison points) results in a bias less than 0.2 ms$^{-1}$ and a standard deviation less than 2.5 ms$^{-1}$ for both horizontal wind components, which we will consider as the final best estimates of the errors of our wind retrieval technique.

## 235    4 Conclusions

In this study we have used dedicated shipborne Doppler cloud radar observations around Darwin, Australia, to evaluate the potential of retrieving vertical profiles of 3D winds using a stabilized platform pointing in successive off-zenith directions at regular intervals. A challenge with using such setup is that the maximum off-zenith angle is 8°, which does not correspond to a large projection of the horizontal wind components onto the radar beam directions. Using this 8° value currently implies that

only ship motions up to 4° in any direction can be compensated for by the stabilized platform. Taking advantage of this "profiler mode" sampling, we have developed a variational 3D wind retrieval technique allowing for 1-minute 3D wind profiles to be estimated. Fully quantitative validation of the results was challenging, as there are no directly collocated observations in time or space available. However, statistical comparisons with radiosonde launches every 3 hours from the ship demonstrated that accurate 3D wind profiles could be derived from such cloud radar observations at small off-zenith angles for a large variety of

cloud cover types encountered during the field experiment, with biases less than 0.2 ms$^{-1}$ and a standard deviation of the errors less than 2.5 ms$^{-1}$. Given the positive results obtained with 8° angles, we will test even lower angles during our next shipborne field experiment. If satisfying results are obtained at even lower angles, this would improve our capability to retrieve 3D winds in much rougher seas than those encountered during the YMCA experiment.

**Acknowledgments**

The Authors wish to thank the CSIRO Marine National Facility (MNF) for its support in the form of *RV Investigator* sea time allocation on Research Voyage IN2019_V06, support personnel, scientific equipment, and data management.

**Code availability**

Codes developed for this study are protected intellectual property of the Bureau of Meteorology and are not publicly available.

**Data availability**

All Doppler cloud radar, radiosonde, and ship underway data are available on the CSIRO Data Access Portal (https://data.csiro.au/dap/).

## Sample availability

No samples were used in this study.

## Author contribution

AP and IM collected the datasets used in this study. IM designed the stabilized platform operating modes. AP analysed the cloud radar and radiosonde observations and wrote and reviewed the manuscript. IM provided edits of the manuscript.

## Competing interests:

The authors declare that they have no conflict of interest.

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

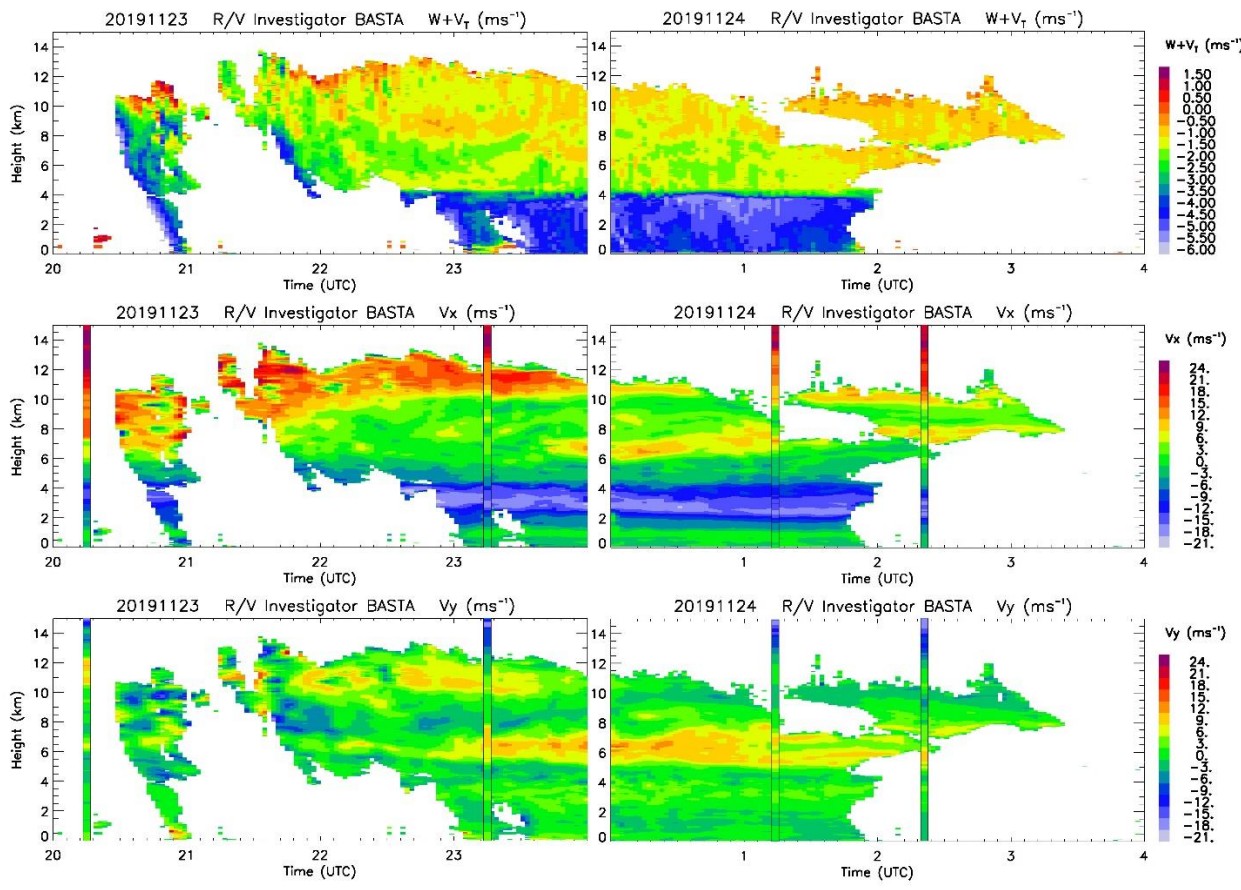


Figure 1: Time – height cross section of retrieved $W+V_T$ (up), $V_X$ (middle), and $V_Y$ (bottom) in a stratiform precipitation case sampled on 23-24/11/2019 by the BASTA cloud radar on *RV Investigator*. Vertical lines on the middle and bottom panels are the horizontal wind components measured by the soundings. The reference time for the soundings is the launch time at ground.

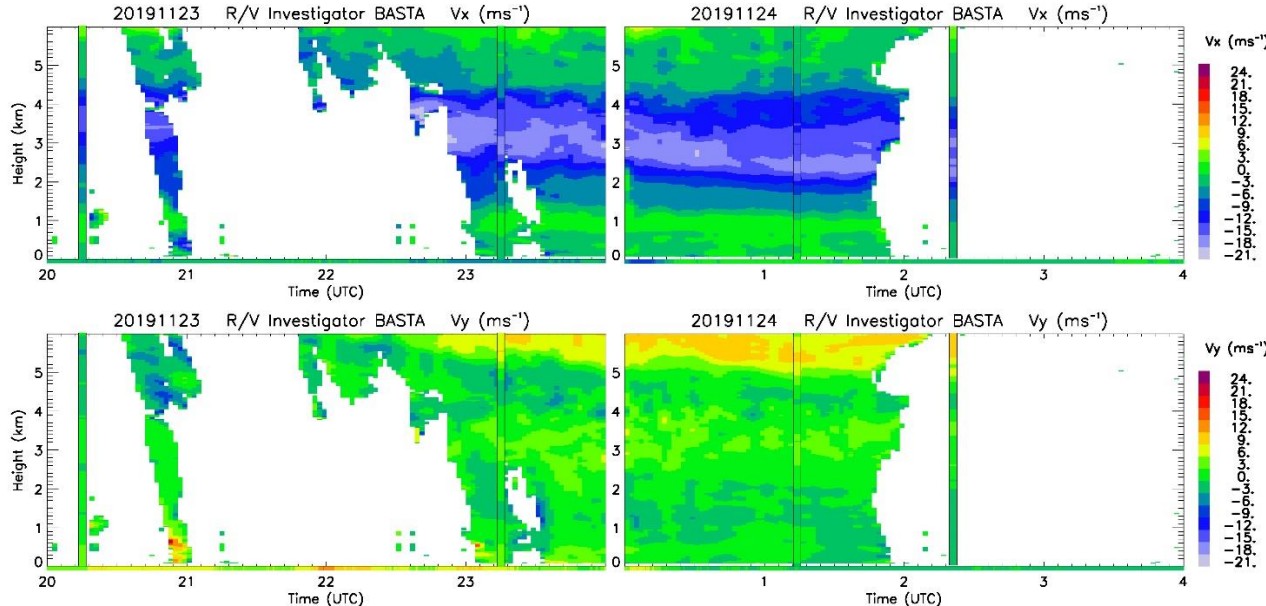


**Figure 2: Time – height cross section of retrieved $V_X$ (top) and $V_Y$ (bottom) in the same case as Fig. 1 but for a maximum altitude of 4 km. The horizontal winds measured on the front mast of *RV Investigator* are displayed at height = 0 with the same colour scale as the retrieved winds.**



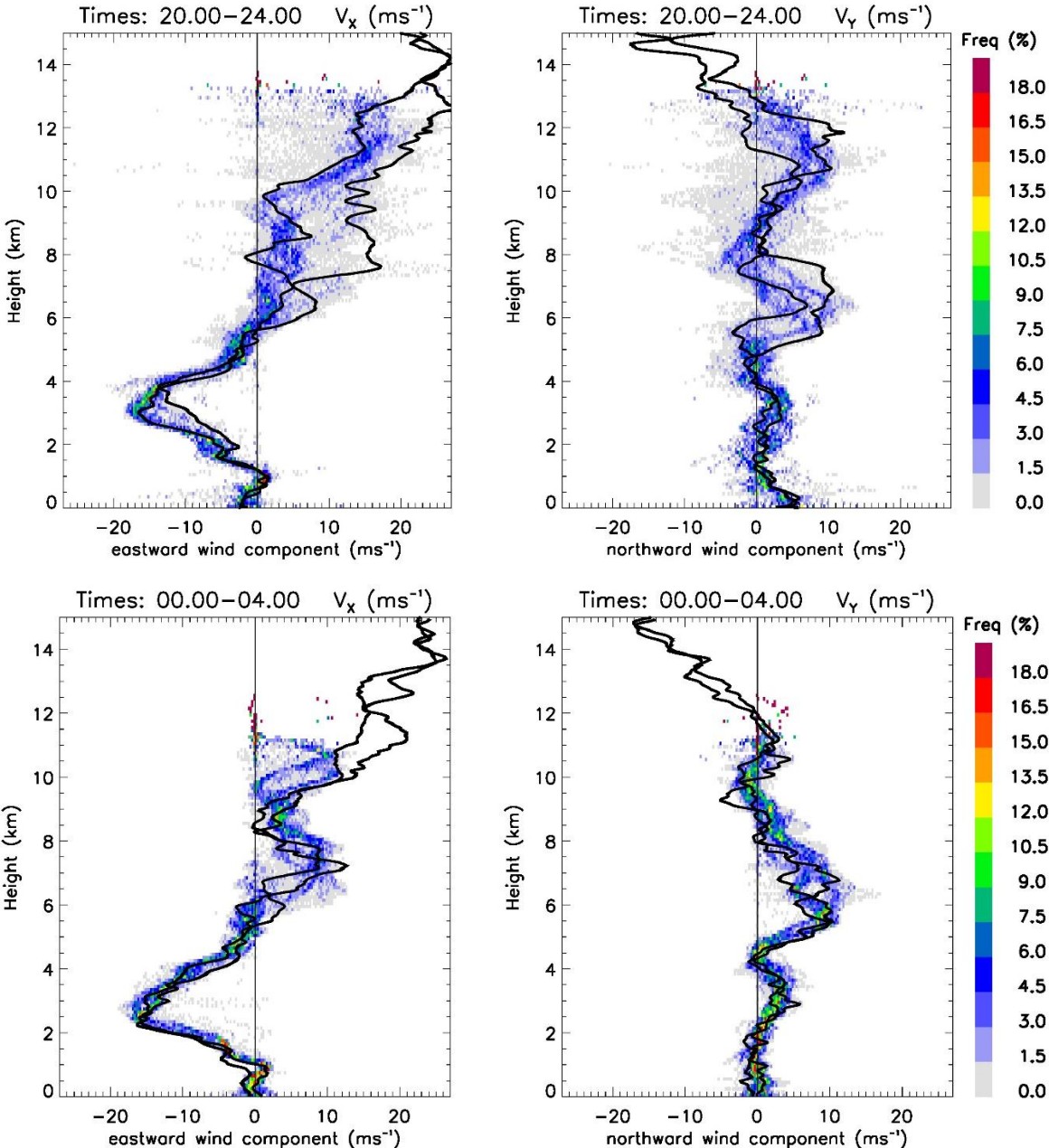

**Figure 3:** Comparison of $V_X$ (left) and $V_Y$ (right) joint wind – height frequency distributions (colours) and the same horizontal wind components measured by soundings for two time periods: 2000-2400 UTC on 23/11/2019 with sounding launches at 2015 and 2315 UTC (top panels) and 0000-0400 UTC on 24/11/2019 with sounding launches at 0115 and 0245 UTC (bottom panels).

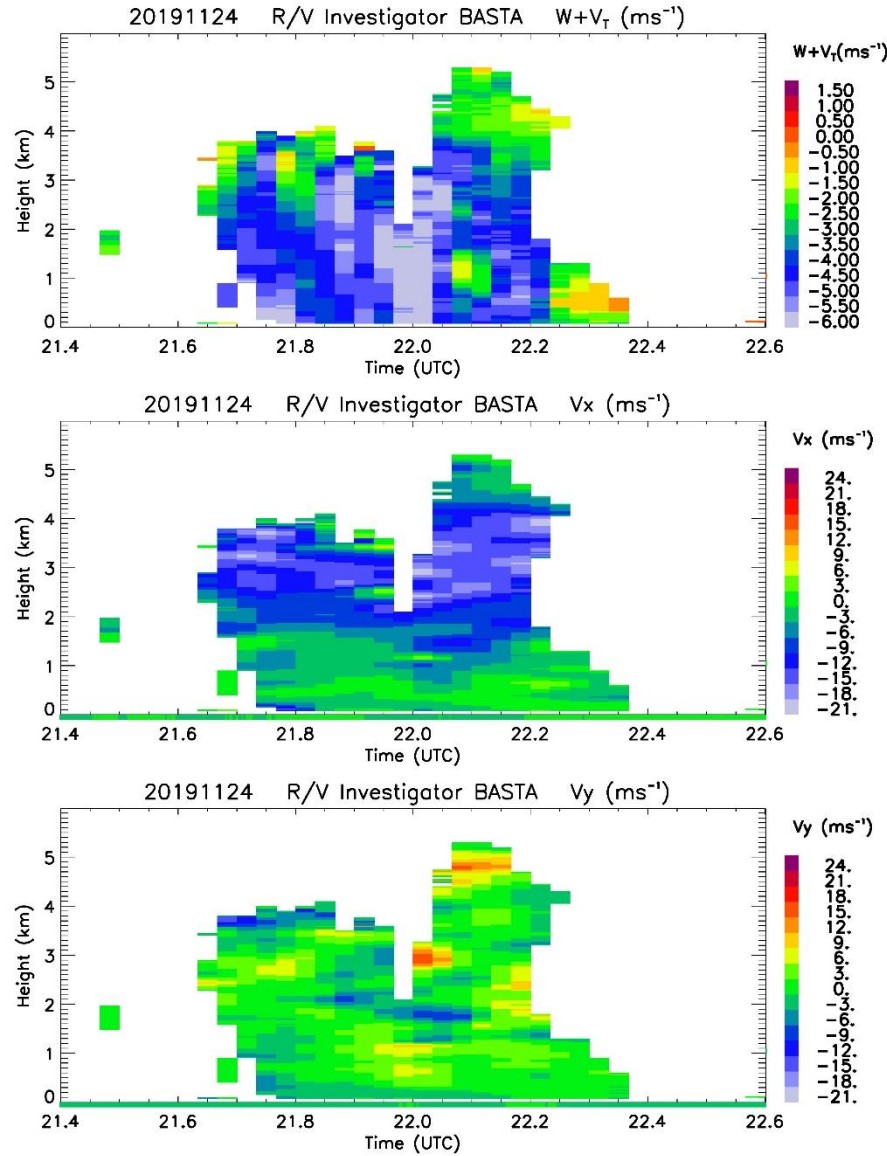

**Figure 4: Same as Fig.2 but for the cumulus congestus case on 24/11/2019 and a maximum display height of 6 km.**


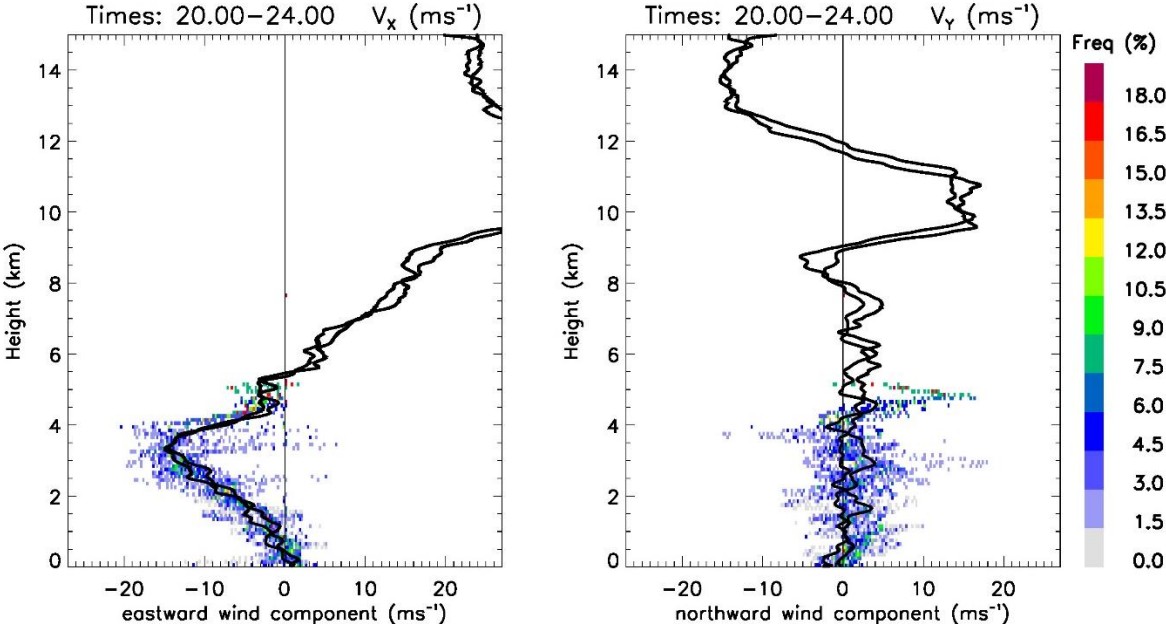


**Figure 5: Same as Fig.3 but for the cumulus congestus case and the 2000-2400 LT period on 24/11/2019.**

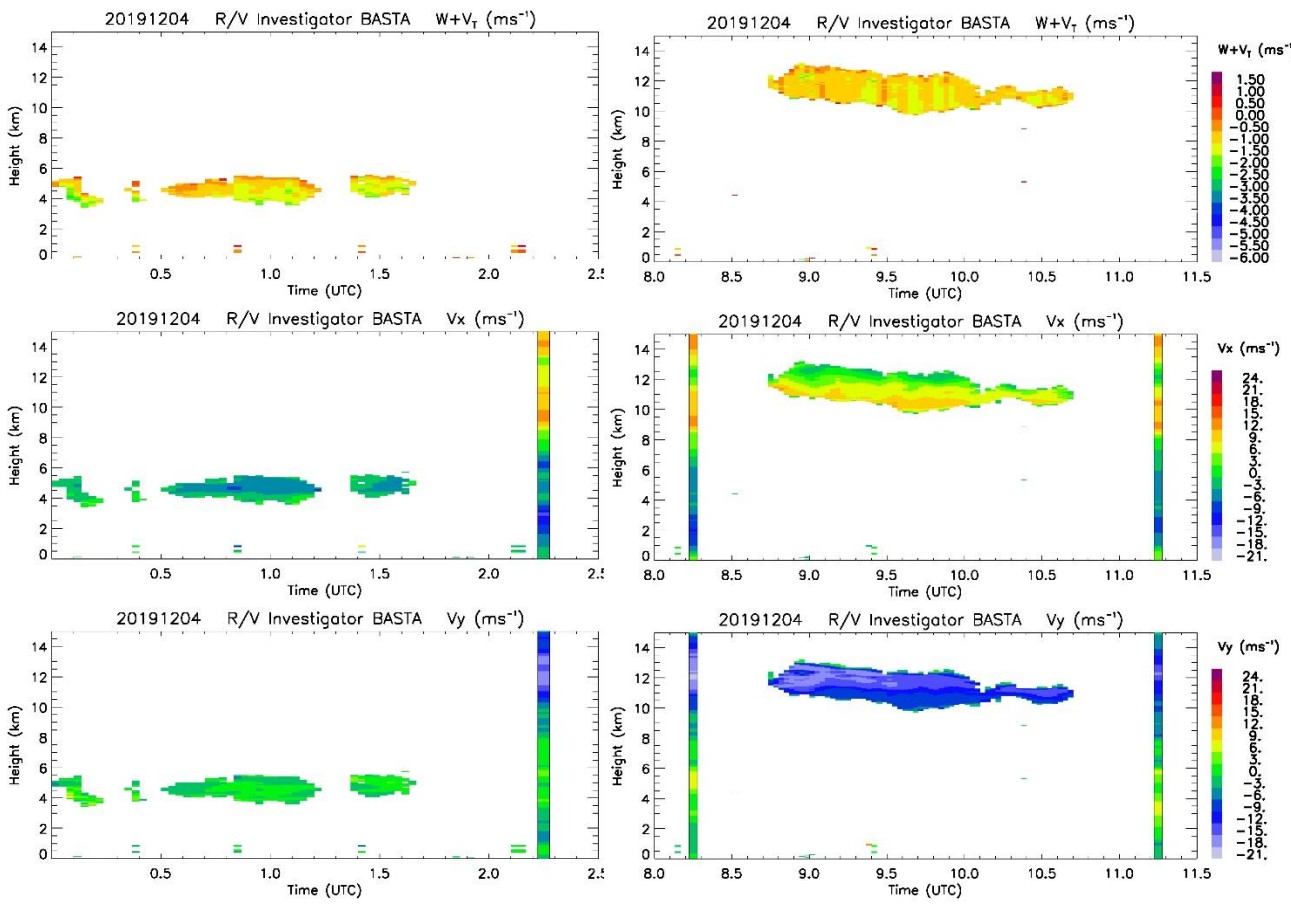


**Figure 6: Same as Fig. 1 but for the altostratus (left) and tropical cirrus (right) cases sampled on 04/12/2019**

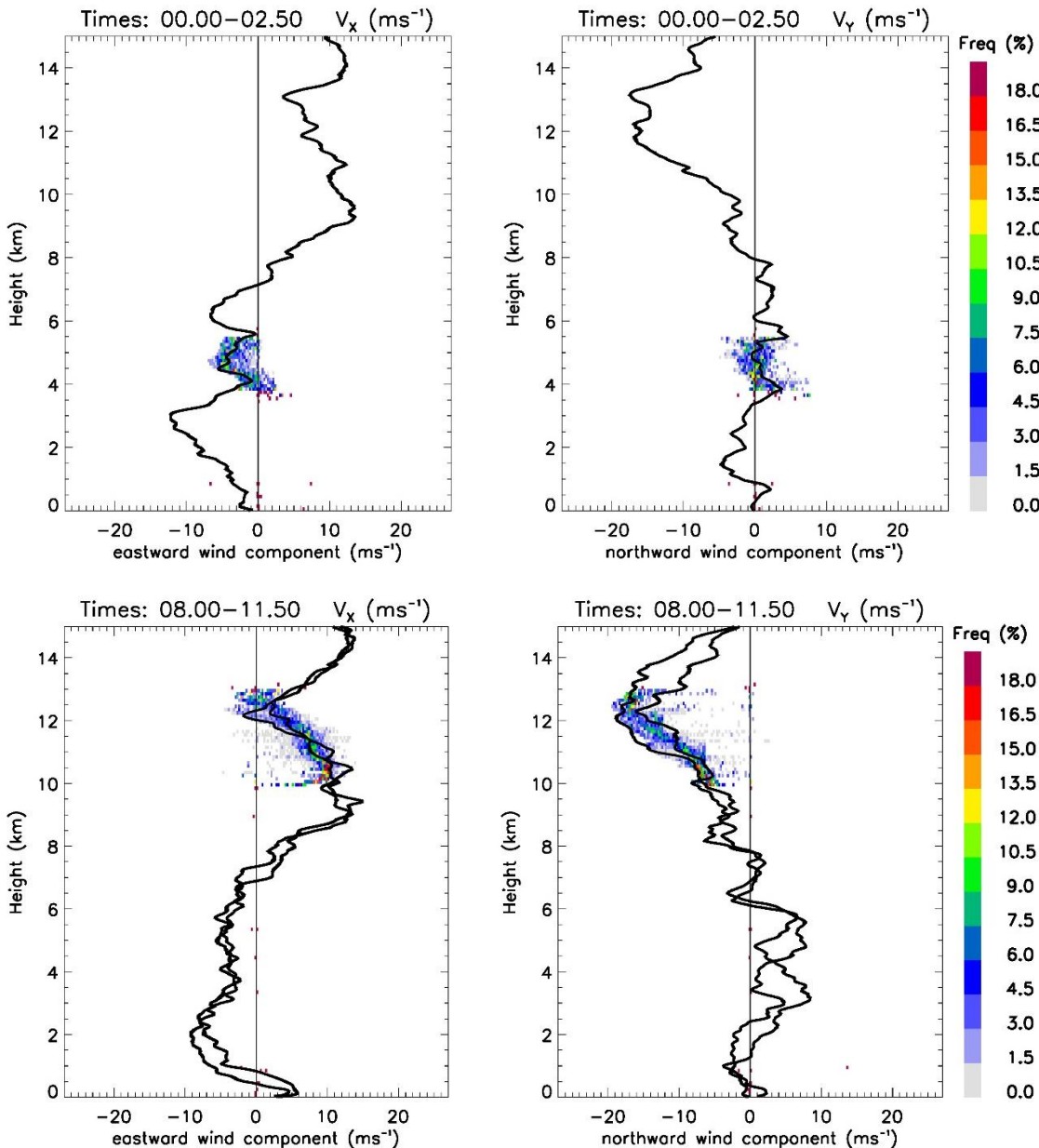


**Figure 7: Same as Fig. 3 but for the altostratus (top panels) and tropical cirrus (bottom panels) cases sampled on 04/12/2019**

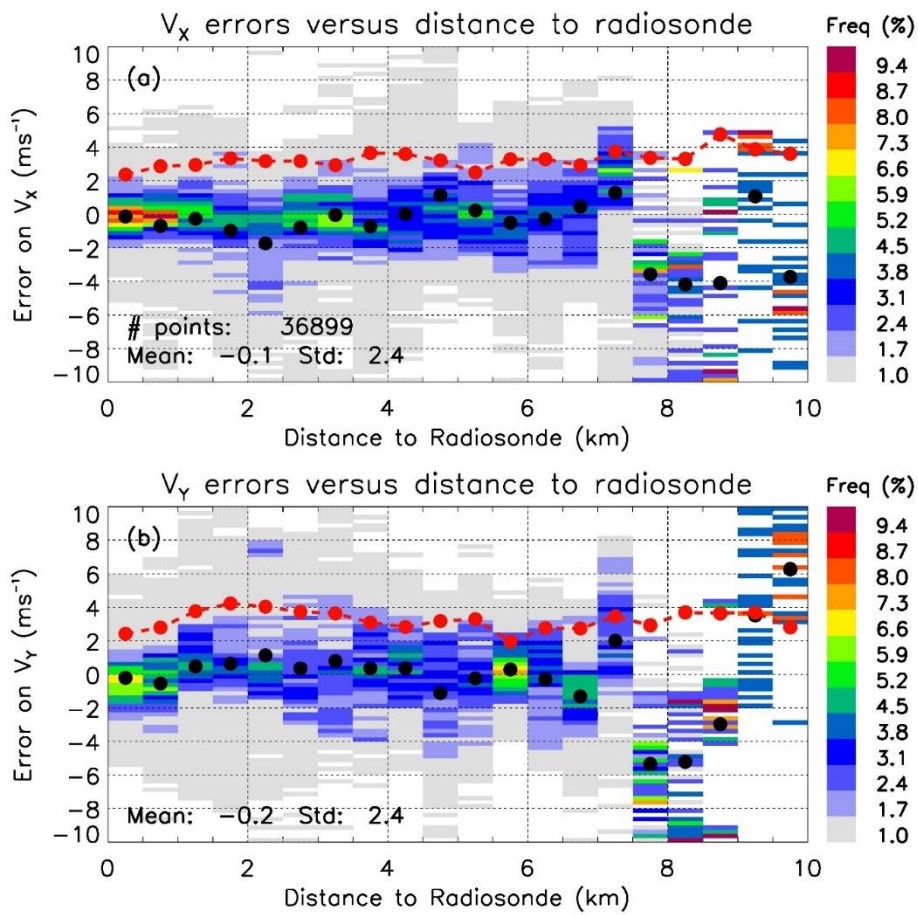


**Figure 8: Frequency distribution of (a) eastward and (northward) wind component errors as a function of the horizontal distance between the ship and the balloon. The sign convention is (retrieved-radiosonde) wind. The frequency is normalized to 100% for each distance bin. Black dots and red dots are the estimates of the bias and standard deviation of the error for each bin, respectively. The mean bias and standard of the error for the first bin is given on each panel. The total number of points is given in panel (a).**


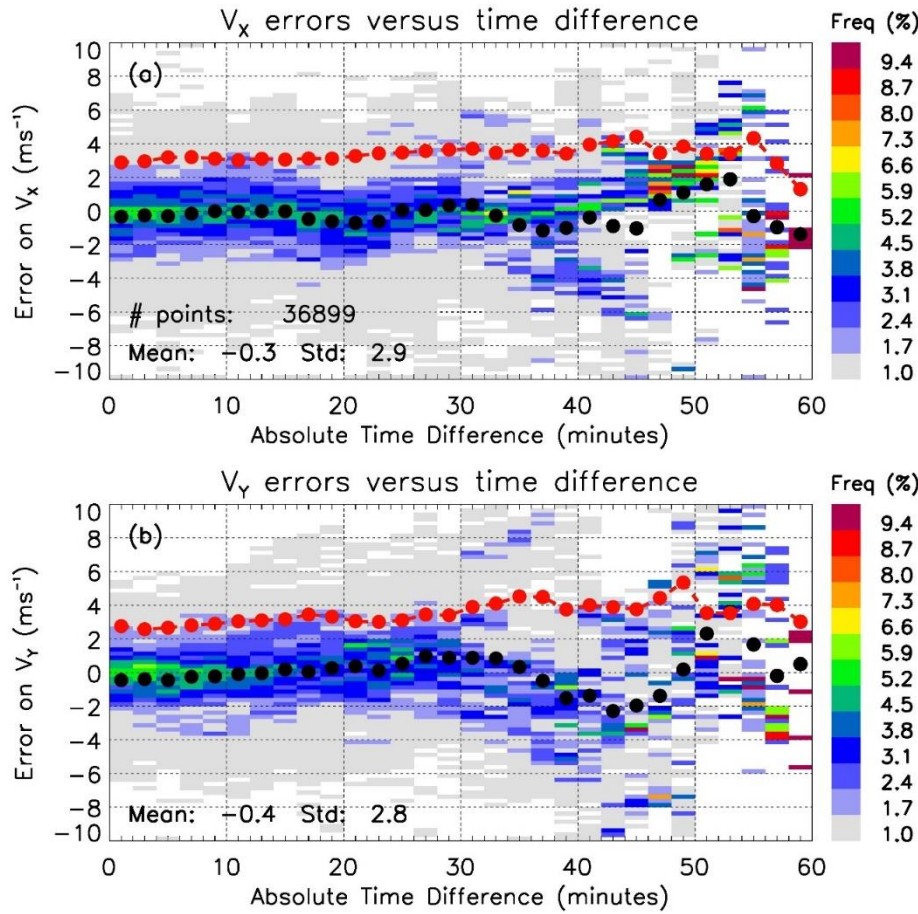

**Figure 9: Same as Figure 8 but using bins of absolute time difference between retrievals and radiosonde measurements.**
