# Peer review of "Three-dimensional wind profiles using a stabilized shipborne cloud radar in wind profiler mode"

_Atmospheric Measurement Techniques, 2020_

## Referee Comment (RC1) · Gerald Mace (Referee) · 25 Feb 2020

This paper describes a method for retrieving the air velocity components from a motion stabalized shipborne millimeter wavelength Doppler radar. The methodology is simplified from a previously published variational methodology that was developed for dual-Doppler observations. The method assumes that the Doppler velocity measurements are stabilized in pitch and roll with ship's heave motion removed from the measurements prior to applying the variational method. The authors explain that the the challenge with the method is that the radome only allows for off-zenith pointing 8 degrees which is much less than typically used. The method is applied to several case

studies observed in the tropics and compared against radiosonde measurements of horizontal wind. The comparison is quite good suggesting that the methodology is successful. The paper is well-written, concise, and informative. It demonstrates an important capability that is new for ship-borne W-Band radar. I suggest publciaton with minor revision.

I would like to see the authors address the following:

1. I think there needs to be a more careful discussion of uncertainty. A close look at the figures suggests that there is a distribution of retrievals at each height with a fairly strong peak. The distribution is particularly noticeable in the 20-24 plots on the top row of Figure 3. Is each of these points a reasonable retrieval or is the distribution caused by noise in the retrievals? Is part of the error budget the precision in the Doppler velocity measurement itself, does the noise arise from the pointing, etc? The congestus highlights the instantaneous aspects of the retrievals whereas the stratiform cases as depicted represent many hours of data yet the same level of noise seems to be present in both. There does seem to be some rather odd spikes in Vy in the congestus example on the sides of the cloud and at the top that do not show up in the vertical or eastward components. Are these real or outliers?

A short discussion regarding these issues would demonstrate how accurate you expect the single retrievals to be and how much averaging is expected to be necessary to converge on a useful solution. It would be interesting to show an actual updraft if such an example is available.

2. It would be nice to include plots of the radar reflectivity in the figures.

The only typographical issue I see is on line 28 where it should read Plan Position Indicator not Plane.

---

## Referee Comment (RC2) · Anonymous Referee #2 · 9 Mar 2020

To understand the quality of this manuscript, the reader only needs to read this one sentence of the paper on p. 6.

"The purpose of the remaining figures of this study is to demonstrate that this good agreement for the stratiform precipitation case holds true for different types of cloud cover, including a cumulus congestus case characterized by high vertical wind shear (Figs. 4 and 5), an altostratus case in a very light wind environment (Figs. 6 and 7), and a tropical cirrus case embedded in a north-westerly jet (Figs. 6 and 7)."

More than half of the figures in the text are explained in this one sentence with the conclusion that these figures represent "good agreement". No further explanation is

provided. No quantitative assessment is provided in the text. The reader is expected to examine the figures for herself and then be convinced that this is good agreement.

In short, high-quality scientific writing is not like this.

Although the authors are clearly working on an important and relevant problem that is appropriate for AMT, weaknesses in the presentation of the paper similar to the issue above detract from what could have been a quality submission.

1. Examine the abstract. There is no quantitative information about the comparison. There is no information about how many soundings are used. The sentences are vague and lack quantification when they should be clearly quantified: "small component of the Doppler velocity in most cases", "Statistical comparisons.... demonstrate that accurate 3D wind profiles can be obtained." There is no information about how much data is evaluated. What about the errors? How is accuracy defined? These are not quantified. How is the comparison performed? What metrics are used? This information is not provided. Thus, the abstract does not serve its role as a concise summary of the manuscript.

2. Paragraphs are not indented, making it difficult to read this manuscript.

3. Line 26: "very little has been done so far": Does this mean that there have been zero studies? If so, say so. If there have been some research, please discuss.

4. The data and code availability statement "upon request to [author]" does not presently adhere to the AMT standards for the data policy: https://www.atmospheric-measurement-techniques.net/about/data_policy.html

5. Line 175: Why four hours? Is this because of radiosonde travel time? If so, please state that to be more clear.

6. I am curious as to why v_x and v_y are analyzed separately. Wouldn't this analysis be sensitive to small errors in the wind direction? You may have better agreement showing wind direction and speed instead.

7. I am not sure whether this paper may be of use to you, but I offer it in case you find it useful. Trapp, R.J. and C.A. Doswell, 2000: Radar Data Objective Analysis. J. Atmos. Oceanic Technol., 17, 105–120.

8. Even with the analysis of Figs. 1–3 at lines 158–182, I feel that not enough is said about the quantitative metrics of the comparison in Fig. 2. How good is it? When combined with Figs. 5 and 7, what are the main results? Is there a mean error that can be quantified? I'm really hoping the authors can provide more quantitative measurements of the quality of the comparisons. Not doing so makes this contribution quite weak scientifically. Yes, you may not have sampled the large number of cases that you had hoped, but more needs to be said about the statistics of the cases you did analyze. Readers of a technical journal such as AMT should expect this information at a minimum.

Other minor concerns:

1. The authors tend to use "which" when they should use "that". Lines 23, 113, 131https://www.grammarly.com/blog/which-vs-that/

2. The authors need to be more careful in the proofreading with the correct use of hyphens versus en dashes. When using an en dash to connect two numbers ("2–4 km" in LaTeX), do not include spaces on either side.

3. Better proofreading is needed. I noticed some examples where commas were omitted.

4. Should the word "resolution" be changed to "data interval" (or similar). If there are data every 15 seconds, you can't resolve features that are 15 seconds. You need 5–8 time steps to resolve a feature.

5. Line 30: Should "in" be "of"?

6. Line 45-46 repeats earlier text.

7. Lines 95–96: The parentheses aren't italicized.

8. Line 99: Insert a comma after "sampling".

9. Lines 106 and 111: "Where" and "With" should not be capitalized., They are continuing from the previous equation. They are not starting new sentences.

10. Line 111: Change "which need" -> "needed".

11. Line 122: Change "since" -> "because".

12. Line 158: "Figs." should be "Figures" because it starts the sentence.

13. Equation 1: I didn't see that i and k were defined. My apologies if I missed it.

14. I kept wondering if the soundings were released from the ship. Could you provide a more clear explanation? If I missed that information in the manuscript, I apologize.

---

## Author Comment (AC1) · 23 Apr 2020

(see attached PDF) - below is a copy of the responses

Reviewer 1: Thanks for the review and for recommending minor revisions. Below we respond point by point to the comments and suggestions.

I would like to see the authors address the following:

1. I think there needs to be a more careful discussion of uncertainty. A close look at the figures suggests that there is a distribution of retrievals at each height with a fairly strong peak. The distribution is particularly noticeable in the 20-24 plots on the top row

of Figure 3. Is each of these points a reasonable retrieval or is the distribution caused by noise in the retrievals?

This is an important point. As explained in the second and third paragraphs of section 3, the main limitation of these comparisons with radiosondes is that an unknown part of the differences is the temporal and spatial variability of the wind components (including internal cloud dynamics at small scale), the other part being the error of the retrieval. One way to get a sense of this spatial and temporal variability is to plot the two successive radiosonde profiles, which span about four hours from the time of launch of the first one to the arrival time of the second in the upper troposphere. Then, for each of these four-hour time intervals, we have produced the distribution of wind retrievals. Our qualitative indication of a "good agreement" is essentially when the distribution of retrievals at each height is bounded by the two radiosonde measurements. When writing this paper, we initially thought that further disentangling those two sources of differences was not possible. However, as it turns out, radiosondes from this experiment usually did not get further away from the ship by more than about 10 km, and from the new quantitative analysis we have conducted (see new version of the manuscript) we have now been able to characterize the errors more quantitatively. To do so, we have binned all comparisons with the time and spatial difference between observations and retrievals. We believe this provides the best possible quantitative assessment of our retrievals given the independent measurements we have.

Is part of the error budget the precision in the Doppler velocity measurement itself, does the noise arise from the pointing, etc? The congestus highlights the instantaneous aspects of the retrievals whereas the stratiform cases as depicted represent many hours of data yet the same level of noise seems to be present in both.

What is referred to as "noise" by the reviewer is actually not noise, it is the variability of the retrieval over four hours, which ideally should be bounded by the two radiosonde profiles, noting that inside the four hours there may be more variability than what has been measured with only two radiosondes. Precision of Doppler measurements is

indeed part of the error, but not expected to be the main one by far, as we do pulse pair over 2048 samples to describe a Nyquist velocity interval of about 10 ms-1, resulting in a Doppler spectral accuracy of 0.6 cms-1.

There does seem to be some rather odd spikes in Vy in the congestus example on the sides of the cloud and at the top that do not show up in the vertical or eastward components. Are these real or outliers?

Well spotted. Those areas on the edges of clouds are where the detection limits of the instrument are reached, so what we see could be a contamination of the retrieval at very low signal-to-noise ratio. However we cannot discard the possibility of local turbulent structures on the edges of the clouds responsible for entrainment and detrainment on these edges. Difficult to tell for sure.

A short discussion regarding these issues would demonstrate how accurate you expect the single retrievals to be and how much averaging is expected to be necessary to converge on a useful solution. It would be interesting to show an actual updraft if such an example is available.

The new quantitative analysis of errors presented in the new version of the manuscript fully addresses this question of individual retrieval accuracy. Our new results show that virtually unbiased wind components are produced (bias less than 0.2 ms-1) with a standard deviation of about 2.5 ms-1 on the horizontal wind components.

2. It would be nice to include plots of the radar reflectivity in the figures.

As radar scientists we can only agree that reflectivity would provide a nice context to the Doppler observations and wind retrievals, however, we feel those plots are not fully needed to discuss the wind retrievals and would result in large four-panel figures in the paper. As a result, unless there is an important reason we have overlooked, we have not added them in the new version of the manuscript.

The only typographical issue I see is on line 28 where it should read Plan Position

Indicator not Plane.

This has been corrected, thanks.

Please also note the supplement to this comment:
https://www.atmos-meas-tech-discuss.net/amt-2020-34/amt-2020-34-AC1-
supplement.pdf
* * *

---

## Author Comment (AC2) · 23 Apr 2020

(see PDF attached) - below copy of the responses.

Reviewer 2:

We would like to thank Reviewer 2 for his comments on the paper, which ultimately guided us to produce a revised version of the manuscript. We must admit we felt a bit puzzled by the main criticism raised in this review, because we felt we had fully disclosed and tried hard to address the limitations stated by the reviewer. However, a more quantitative analysis presented in the new version of the manuscript proved that

our initial assumption that we could not produce a quantitative estimate of the errors was wrong. We have addressed the other comments thoroughly in our revised version and hope the reviewer will now be satisfied with the quality of our study.

As we explained in the initial manuscript, the reason why we thought we could not do a full quantitative estimation of the errors of our retrievals is that the radiosondes drift away quickly from the ship location with time as it ascends. So, differences between retrievals and independent measurements cannot be attributed to errors of the retrieval alone. It can be due to the unknown spatial and temporal variability of the 3D wind components (including the internal cloud dynamics). As a result, although we were as frustrated as the reviewer about the inability to fully quantify errors in our retrievals, we didn't think the measurements we have allowed for such quantification. The rule we came up with to make best use of our observations and label a comparison a "good agreement" was to use (as we try to explain more clearly in section 3) sets of two radiosonde launches bounding periods of horizontal wind retrieval profiles, and to produce height-dependent PDFs of horizontal wind retrievals between those two times, then to compare this spread of the retrievals with that measured by the two radiosondes. If the 4-hour PDFs of horizontal wind retrieval fell between the two radiosonde measurements, we labelled the comparison as a good agreement, because the retrievals are within the natural variability as captured by the two soundings.

These four-hour case studies did not provide a quantitative evaluation of the retrieved horizontal winds, which is the main criticism of the reviewer. As explained earlier, the main challenge of these comparisons with soundings is that differences include two components: the errors from the retrieval, which is what we would like to characterize, and the spatial and temporal variability of the true horizontal wind components captured by the soundings as the balloons drift away from the ship location during the hour it takes from the balloon to go from ground to the tropopause. Despite these challenges, the second term should vanish when the spatial distance and the time difference between the retrieved and measured horizontal winds is minimal. In order to investigate

these effects and quantify errors in our horizontal wind estimates as accurately as possible, all comparison points have been binned as a function of distance (every 0.5 km from 0 to 10 km) and absolute time difference (every 2 minutes from 0 to 60 minutes) between sounding and retrievals. It turned out to be much more informative than what we initially assumed, and we now think that we have produced a quantitative estimate of the retrieval errors using the shortest distance and absolute time differences. These results, presented in new figures 8 and 9, are now fully presented and described in the new version of the manuscript. More specifically, the following text has been written :

" The case studies presented previously do not provide a quantitative evaluation of the retrieved horizontal winds. As explained earlier, the main challenge of these comparisons with soundings is that differences include two components: the errors from the retrieval, which is what we would like to characterize, and the spatial and temporal variability of the true horizontal wind components captured by the soundings as the balloons drift away from the ship location during the hour it takes from the balloon to go from ground to the tropopause. Despite these challenges, the second term should vanish when the spatial distance and the time difference between the retrieved and measured horizontal winds is minimal. In order to investigate these effects and quantify errors in our horizontal wind estimates as accurately as possible, all comparison points are binned as a function of distance (every 0.5 km from 0 to 10 km) and absolute time difference (every 2 minutes from 0 to 60 minutes) between sounding and retrievals. The statistical frequency distribution of errors (normalized to 100% in each bin), bias, and standard deviation of the differences are presented in Fig. 8 for distance and Fig. 9 for absolute time difference. Fig. 8 indicates that quantitative comparisons can be made between soundings and retrievals up to about 7 km distance (with fluctuations of the bias and standard deviation of about 1 and 1.5 ms-1, respectively). However, it appears clearly that the frequency distribution of errors is more peaked for distances below 1km, showing as expected some impact at relatively short distance of the spatial variability on the accuracy of our error estimates. The statistical analysis of all comparison points at distances less than 0.5 km (6.4% of all points), which is our best estimate

of the retrieval errors since they include the smallest possible contribution from the true spatial variability, indicate that virtually unbiased (less than 0.2 ms-1) estimates of the two horizontal wind components are obtained, with a standard deviation of the error of 2.4 ms-1. Reorganizing the comparison points using bins of absolute time difference between soundings and retrieval points (Fig. 9) yield similar results to those obtained when binning using distance. From the samples used to produce Fig. 9, we obtain that the temporal evolution of the horizontal wind components has a large impact on our error estimates beyond time differences of 30 minutes, but a reasonably small effect on our error estimates for absolute time differences up to about 15-20 minutes (about 1 ms-1 for the bias and standard deviation of the error), with the distribution of errors getting broader for absolute time differences greater than about 15-20 minutes. As was observed for the analysis as a function of distance, the frequency distribution of errors is much more peaked for times less than about 6 minutes, which is therefore where we expect minimal contamination of our error estimates due to temporal variability. It is interesting to see that the frequency distribution for small distances is much more peaked than that for short time differences. This suggests that the temporal variability of the horizontal wind components has some impact on our error estimates than the spatial variability. The statistical analysis of all comparison points with less than 2 minutes of absolute time difference (8.5% of all points) confirms the small bias of the two horizontal wind components (less than 0.4 ms-1) that was obtained using distances smaller than 0.5 km, and a similar but slightly higher standard deviation of the error (2.9 ms-1). Using both distances less than 0.5 km and absolute time differences less than 2 minutes (3% of all comparison points) results in a bias less than 0.2 ms-1 and a standard deviation less than 2.5 ms-1 for both horizontal wind components, which we will consider as the final best estimates of the errors of our wind retrieval technique."

This response above covers most of the comments raised by the reviewer, and the new quantification of errors presented fully addresses this reviewer's concerns. However, below, we provide a point-by point response with more details when appropriate.

To understand the quality of this manuscript, the reader only needs to read this one sentence of the paper on p. 6. "The purpose of the remaining figures of this study is to demonstrate that this good agreement for the stratiform precipitation case holds true for different types of cloud cover, including a cumulus congestus case characterized by high vertical wind shear (Figs. 4 and 5), an altostratus case in a very light wind environment (Figs. 6 and 7), and a tropical cirrus case embedded in a north-westerly jet (Figs. 6 and 7). " More than half of the figures in the text are explained in this one sentence with the conclusion that these figures represent "good agreement". No further explanation is provided. No quantitative assessment is provided in the text. The reader is expected to examine the figures for herself and then be convinced that this is good agreement. In short, high-quality scientific writing is not like this.

That is because, as we explained, we had established a rule to label a good agreement (see main response above) and felt there was not much more that could be said from these figures. The wind component PDFs are or are not within the variability measured by the radiosondes. The most important point we address visually with these plots is whether the retrieval behaves sensibly in very different types of clouds. However, we agree that this was a bit dry, and there were a few more things that needed to be highlighted on these figures, so, in order to address this comment, we have added the following text : " Comparisons with ship-level weather stations and with the radiosondes bounding the congestus case (Figs. 4 and 5) indicate that the near-surface winds are well reproduced and the strong vertical wind shear in the low levels (about 5 10-3 s-1 in the 0-3 km layer) is accurately captured by the retrieval. We note some large differences with the radiosonde on the VY component at cloud top, which does not necessarily mean that the retrieval has failed, as tops of congestus clouds are notoriously turbulent. Therefore, these differences could be due to the internal convective-scale dynamics not captured by soundings obtained in clear-air. The altocumulus and cirrus cases are characterized by a much lower temporal variability of the wind components (see for instance the small difference between estimates from the two soundings for the cirrus case, Fig. 7). Accordingly, the distributions of retrieved horizontal winds over four

hours is much narrower than within the other cases, and the peaks of the distributions align well with the radiosonde measurements for those two cases."

Although the authors are clearly working on an important and relevant problem that is appropriate for AMT, weaknesses in the presentation of the paper similar to the issue above detract from what could have been a quality submission.

1. Examine the abstract. There is no quantitative information about the comparison. There is no information about how many soundings are used. The sentences are vague and lack quantification when they should be clearly quantified: "small component of the Doppler velocity in most cases", "Statistical comparisons.... demonstrate that accurate 3D wind profiles can be obtained." There is no information about how much data is evaluated. What about the errors? How is accuracy defined? These are not quantified. How is the comparison performed? What metrics are used? This information is not provided. Thus, the abstract does not serve its role as a concise summary of the manuscript. There is no information about how much data is evaluated. What about the errors? How is accuracy defined? These are not quantified. How is the comparison performed? What metrics are used?

See main response above. This is now comprehensively done in the abstract, section 3, and conclusion.

2. Paragraphs are not indented, making it difficult to read this manuscript.

Thanks for pointing that out. We have now indented the paragraphs throughout.

3. Line 26: "very little has been done so far": Does this mean that there have been zero studies? If so, say so. If there have been some research, please discuss.

It was indeed a vague statement, so we have reworded as " The next challenge to better understand these interactions between dynamics and cloud microphysics is to characterize the dynamical context of these cloud microphysical observations, including horizontal winds, vertical wind shear, and entrainment processes within and at the

boundaries of clouds."

4. The data and code availability statement "upon request to [author]" does not presently adhere to the AMT standards for the data policy: https://www.atmosphericmeasurement- techniques.net/about/data_policy.html

Thanks for pointing that out. Regarding code availability, we are only "encouraged" to deposit software, but I don't plan to do that due to protection of the Bureau IP on this (a version of this wind retrieval technique is currently being used to develop operational products that are planned to be sold commercially), so I rephrased as : "Code developed as part of this study is not available publicly as it is intellectual property of the Bureau of Meteorology." I just looked at the data policy and searched "CSIRO data access portal" in the r3data.org, and it is there. The datasets used in this study are now posted on the CSIRO DAP, so I wrote this in the section :" All data and wind retrievals used in this study are publicly available on the CSIRO Data Access Portal (https://data.csiro.au/dap/)."

5. Line 175: Why four hours? Is this because of radiosonde travel time? If so, please state that to be more clear.

This information has been added. From the release time of the first radiosonde and the arrival time of the second radiosonde there are indeed about four hours, which is why we use this time span for each comparison.

6. I am curious as to why v_x and v_y are analyzed separately. Wouldn't this analysis be sensitive to small errors in the wind direction? You may have better agreement showing wind direction and speed instead.

There is value in using either the two wind components or the wind and direction, we agree. Using the two wind components avoids issues with 360 degrees flips of the wind direction, so we have kept the option of using the two horizontal wind components. Besides, this is formally what is retrieved.

7. I am not sure whether this paper may be of use to you, but I offer it in case you find it useful. Trapp, R.J. and C.A. Doswell, 2000: Radar Data Objective Analysis. J. Atmos. Oceanic Technol., 17, 105–120.

This very interesting paper (which is a very important one for radar scientists like us) is not relevant for this study, as we are not interpolating volumetric radar observations onto a Cartesian grid at all.

8. Even with the analysis of Figs. 1–3 at lines 158–182, I feel that not enough is said about the quantitative metrics of the comparison in Fig. 2. How good is it? When combined with Figs. 5 and 7, what are the main results? Is there a mean error that can be quantified? I'm really hoping the authors can provide more quantitative measurements of the quality of the comparisons. Not doing so makes this contribution quite weak scientifically. Yes, you may not have sampled the large number of cases that you had hoped, but more needs to be said about the statistics of the cases you did analyze. Readers of a technical journal such as AMT should expect this information at a minimum.

See detailed response above. Also, at the time our problem was never about the number of cases, it was about the limitations of the potential validation datasets, as explained. We hope we have now convinced the reviewer about this important point. New results included also fully addressed this comment.

Other minor concerns:

1. The authors tend to use "which" when they should use "that". Lines 23, 113, 13 1https://www.grammarly.com/blog/which-vs-that/

Thanks for the link. We have reviewed our use of "which" versus "that" using these explanations. We made use of the term "which" 13 times in the initial manuscript. Of these 13 times, we indeed used "which" 4 times in defining clauses. This has been corrected.

[Figure]

2. The authors need to be more careful in the proofreading with the correct use of hyphens versus en dashes. When using an en dash to connect two numbers ("2–4 km" in LaTeX), do not include spaces on either side.

This has been corrected.

3. Better proofreading is needed. I noticed some examples where commas were omitted.

We have carefully proof-read the new version of the manuscript, with attention paid to omitted commas.

4. Should the word "resolution" be changed to "data interval" (or similar). If there are data every 15 seconds, you can't resolve features that are 15 seconds. You need 5–8 time steps to resolve a feature.

We agree that this is formally an improper use of the word "resolution". We worked around this as follows: " The selected time interval of 15 seconds is a trade-off to make sure that we are collecting data from all four radar modes, which takes 12 seconds, for each pointing direction while still retaining a small time interval (1 minute) between retrieved 3D wind profiles."

5. Line 30: Should "in" be "of"?

I think both are acceptable in this context but we will go with your suggestion.

6. Line 45-46 repeats earlier text.

We have removed the first part of this short paragraph to still provide a short introduction to the section.

7. Lines 95–96: The parentheses aren't italicized.

Maybe they don't appear as italicized in the PDF conversion, but they are italicized in our Word version, we've double checked.

8. Line 99: Insert a comma after "sampling".

Done.

9. Lines 106 and 111: "Where" and "With" should not be capitalized. They are continuing from the previous equation. They are not starting new sentences.

Corrected.

10. Line 111: Change "which need" -> "needed".

This has been rewritten as " matched to the observed Doppler velocities {\ \mathbit{V}}_\mathbit{R}\ \left(\mathbit{i},\mathbit{k}\right) as part of the minimization process"

11. Line 122: Change "since" -> "because".

Corrected.

12. Line 158: "Figs." should be "Figures" because it starts the sentence.

Corrected.

13. Equation 1: I didn't see that i and k were defined. My apologies if I missed it.

They are just the indices in the summations used in Equation 1 and because they go from 0 and nt and 0 and nz respectively, it is clear that these indices refer to summations over the times and range bins. I don't think it is customary to define the indices in this case (?).

14. I kept wondering if the soundings were released from the ship. Could you provide a more clear explanation? If I missed that information in the manuscript, I apologize.

Sorry, we thought it was obvious. We added "from the ship" in four different locations to make sure there is no ambiguity about that.

Please also note the supplement to this comment:

https://www.atmos-meas-tech-discuss.net/amt-2020-34/amt-2020-34-AC2-supplement.pdf

---

## Referee Report (RR1)

The Authors have addressed all of my concerns with the original manuscript. The revised manuscript is ready for publication